# Impact of Cesarean Delivery and Breastfeeding on Secretory Immunoglobulin A in the Infant Gut Is Mediated by Gut Microbiota and Metabolites

**DOI:** 10.3390/metabo13020148

**Published:** 2023-01-18

**Authors:** Yuan Yao Chen, Hein M. Tun, Catherine J. Field, Piushkumar J. Mandhane, Theo J. Moraes, Elinor Simons, Stuart E. Turvey, Padmaja Subbarao, James A. Scott, Anita L. Kozyrskyj

**Affiliations:** 1Department of Pediatrics, Faculty of Medicine and Dentistry, University of Alberta, Edmonton, AB T6G 1C9, Canada; 2The Jockey Club School of Public Health and Primary Care, The Chinese University of Hong Kong, Hong Kong, China; 3Department of Agricultural, Food and Nutritional Science, University of Alberta, Edmonton, AB T6G 2E1, Canada; 4Department of Pediatrics and Physiology, Hospital for Sick Children, University of Toronto, Toronto, ON M5G 1X8, Canada; 5Department of Pediatrics and Child Health, University of Manitoba, Winnipeg, MB R3A 1S1, Canada; 6Department of Pediatrics, BC Children’s Hospital, University of British Columbia, Vancouver, BC V6H 0B3, Canada; 7Dalla Lana School of Public Health, University of Toronto, Toronto, ON M5T 3M7, Canada

**Keywords:** birth mode, infant gut microbiota, metabolites, SIgA

## Abstract

How gut immunity in early life is shaped by birth in relation to delivery mode, intrapartum antibiotic prophylaxis (IAP) and labor remains undetermined. We aimed to address this gap with a study of secretory Immunoglobulin A (SIgA) in the infant gut that also tested SIgA-stimulating pathways mediated by gut microbiota and metabolites. Among 1017 Canadian full-term infants, gut microbiota of fecal samples collected at 3 and 12 months were profiled using 16S rRNA sequencing; *C. difficile* was quantified by qPCR; fecal metabolites and SIgA levels were measured by NMR and SIgA enzyme-linked immunosorbent assay, respectively. We assessed the putative causal relationships from birth events to gut microbiota and metabolites, and ultimately to SIgA, in statistical sequential mediation models, adjusted for maternal gravida status in 551 infants. As birth mode influences the ability to breastfeed, the statistical mediating role of breastfeeding status and milk metabolites was also evaluated. Relative to vaginal birth without maternal IAP, cesarean section (CS) after labor was associated with reduced infant gut SIgA levels at 3 months (6.27 vs. 4.85 mg/g feces, *p* < 0.05); this association was sequentially mediated through gut microbiota and metabolites of microbial or milk origin. Mediating gut microbiota included Enterobacteriaceae, *C. difficile*, and *Streptococcus*. The milk or microbial metabolites in CS-SIgA mediating pathways were galactose, fucose, GABA, choline, lactate, pyruvate and 1,2-propanediol. This cohort study documented the impact of birth on infant gut mucosal SIgA. It is the first to characterize gut microbe-metabolite mediated pathways for early-life SIgA maturation, pathways that require experimental verification.

## 1. Introduction

The most abundant antibody secreted by gut mucosa, secretory Immunoglobulin A (SIgA) represents a first-line immune defense against microbial infection [1,2]. The gut mucosal immune system is poorly developed after birth and relies on breastfeeding as a supply of SIgA to supplement the newborn’s initially low production [3,4]. Indeed, breast milk-derived SIgA enhances gut barrier function in suckling neonates, preventing infection by pathogens [5,6]. Secretory IgA has come to the forefront as a critical molecule in the immune system’s first response to SARS-CoV-2 virus [7,8]. Anti-spike IgA is detected in the saliva of newborns of mothers with COVID-19 [9] and in blood; its levels are higher in breastfed infants [10]. An enhanced antibody response during breastfeeding is also observed against haemophilus influenza vaccine [11].

Early infancy is a critical period for the infant gut to be progressively colonized with resident microbiota that collectively drive gut maturation and immunity [12]. Gut microbiota and their metabolites, as well as dietary compounds, are essential for the priming of immune responses [13,14,15]. While SIgA production is induced by pathogenic microbes and other antigens, gut microbiota and their metabolites are also stimuli [15,16,17]. A balanced gut microbiota during infancy promotes beneficial SIgA responses [18,19]. Cesarean section (CS) delivery disturbs this balance, elevating several Proteobacterial species and depleting *Bifidobacterium* and *Bacteroides*, and raising risk of *C. difficile* colonization [20]. Intrapartum antibiotics prophlaxis (IAP) too can cause gut dysbiosis [21,22]. Whether labor is present or absent with CS also leads to differences in *Bifidobacterium* quantity in infant gut microbiota [23]. Further, vaginally-delivered infants are more likely to be breastfed [24,25]. Many of these birth-related changes have been associated with lowered intestinal SIgA and risk of atopic sensitization [26,27]. There is evidence of biologic pathways involving Proteobacteria that can induce IgA responses that regulate gut microbial maturation [28]; microbial metabolites, lactate and pyruvate, can enhance immune responses [29]. However, knowing how birth and subsequent feeding practices affect endogenous gut SIgA production in infants remains unknown. In this study, we determined the impact of birth mode and breastfeeding status on fecal SIgA levels of infants at 3 and 12 months. Secondly, we identified potential mediating pathways involving infant gut microbiota and metabolites.

## 2. Methods

### 2.1. Study Design

This study of 1017 full-term infants represents a subset of Canadian Healthy Infant Longitudinal Development (CHILD) birth cohort (www.childstudy.ca (accessed on 1 April 2022)). Mothers of studied infants were enrolled in their second or third trimester of pregnancy from Vancouver, Edmonton, Winnipeg study sites between 1 January 2009, and 31 December 2012. They were retained in the study if their newborns were a singleton live birth at ≥35 weeks of gestation with a birth weight greater than 2500 g. In vitro fertilized births were excluded to avoid multiple gestation or preterm births, and home birth were also excluded due to the lack of documentation of maternal IAP. Hospital birth records provided data on infant sex, delivery mode, duration of the first and second stage of labor, maternal IAP and gravida status. Women colonized with vaginal *Group B Streptococcus* (GBS) or delivered by caesarean section were administrated by intrapartum penicillin or cefazolin, respectively. This information was used in this study to categorize infants into respective groups as follows: vaginal-no IAP (vaginal delivery without IAP), vaginal-IAP (vaginal delivery with IAP), CS-labor (labor phase occurred before CS delivery) and CS-no labor (scheduled CS delivery without labor phase). Information on maternal characteristics, such as ethnicity, age, depressive symptom, household pet exposure, education level, prenatal smoking and locations, as well as the infant characteristics, including gestational age, breastfeeding exclusivity and infant antibiotic use before 3 months and breastfeeding duration at 1 year were reported by standardized questionnaires completed by parents. Breastfeeding status at 3 months was categorized into exclusive-BF (breastmilk only), partial-BF (breastmilk supplemented with formula) or non-BF (no longer breastfed). Breastfeeding duration at 1 year was classified as <3 month-BF (ceased breastfeeding before 3 months), 3 to <12 month-BF (ceased breastfeeding between 3 and 12 months) and ≥12 month-BF (breastfeeding greater than 12 months). The demographic characteristics of the infants and their mothers were compared according to different birth events (Appendix A). This study was approved by the Human Research Ethics Boards at the Universities of Alberta, Manitoba and British Columbia.

#### Infant Fecal Microbiota, Metabolites and SIgA

Infant fecal samples collected at 3 months (mean age, 3.7 months) and 12 months (mean age, 12.0 months) were utilized to evaluate: gut microbiota composition, as characterized by diversity metrics and taxonomic abundance from Illumina MiSeq sequencing of V4 regional 16S ribosomal RNA (rRNA) processed by QIIME (v1.6.0) [30]; *C. difficile* colonization detected by qPCR [26]; metabolite concentrations, quantified as micromoles per gram of feces by Nuclear Magnetic Resonance (NMR) spectroscopy [31]; and SIgA levels, quantified in milligrams per gram of wet-weight feces by SIgA enzyme-linked immunosorbent assay [32]. SIgA data were available for 1017 infants at 3 months and 175 infants at 12 months. A complete dataset, characterizing gut microbiota, metabolites and SIgA, for 551 and 175 infants at 3 months and 12 months, respectively, was available for mediation analysis.

### 2.2. Statistical Analysis

Statistical analyses were conducted using R (version 3.5.1, R Core Team, 2020) within RStudio (version 1.2.1335, RStudio Team, Boston, MA, USA, 2020). Maternal and infant characteristics were compared with Fisher’s exact tests. Linear regression modeling examined the association between birth events, or breastfeeding status and fecal SIgA levels. Values of fecal SIgA were Box-Cox transformed before linear regression to improve the normality of data [33]. Three-month and 12-month microbial taxa with a relative abundance > 0.05% were used for downstream analysis. Microbial diversity was estimated by the richness index (Chao1) and alpha-diversity indices (PD whole tree phylogenic diversity and Shannon), then compared across birth event or breastfeeding status groups using the nonparametric Kruskal-Wallis test, followed by Dunn’s post-test for multiple comparisons. Beta-diversity (whole microbial community differences by group) was determined by Bray-Curtis dissimilarity and represented by a PCoA plot; group differences were tested by multivariate ANOVA (Adonis PERMANOVA) with 1000 permutations in R (https://CRAN.R-project.org/package=vegan (accessed on 1 April 2022)). Differential abundance of microbial taxa by group was visualized by stack bar and tested using linear discriminant analysis (LDA) effect size (LEfSe) [34]. Statistical significance was determined by an LDA score > 2 and *p*-value < 0.05, using vaginal-no IAP as the reference group for birth events, exclusive-BF and ≥12 month-BF as the reference group for 3-month and 12-month breastfeeding status, respectively. Using the same reference groups, *C. difficile* colonization rates by birth event were examined by Fisher’s exact test. Spearman’s rank correlations were performed between SIgA and gut microbial abundance following stratification by birth events and breastfeeding status. Microbial taxa found to be correlated with SIgA were further tested for their correlation with metabolite concentrations. Spearman correlation was conducted using the ‘corr.test’ function in R (https://cran.r-project.org/package=psych (accessed on 1 April 2022)). *p* values were corrected for multiple testing (false discovery rates, FDR) using Benjamini–Hochberg adjustment [26], with R “FSA” package [35]. Statistically significant correlations (FDR < 0.05) were displayed by means of a heatmap.

A schematic diagram of the sequential mediation model is shown in Appendix A; the analysis was conducted using the Hayes PROCESS v3.5 macro in SPSS (version 26, SPSS Inc., Chicago, IL, USA) [36]. A multiple mediator path model was performed to determine the indirect effects of sequential mediators: gut microbial taxa acted as mediator 1 and metabolites as mediator 2 in the path between birth events (X) and SIgA (Y). Breastfeeding status, tested as the rank-order variable (non-BF, partial-BF to exclusive-BF) to represent increased dose of breastmilk, also acted as mediator 1. Vaginal-no IAP was the reference group for the birth events variable. The indirect effects of gut microbial taxa and metabolites were also tested for paths between breastfeeding status and SIgA, in which case, exclusive-BF and ≥12 month-BF were reference groups for 3- and 12-month breastfeeding status, respectively. In mediation analysis, the procedure of bootstrapping (5000 bootstrap resamples) was utilized to generate 95% confidence intervals.

### 2.3. Covariates

Potential covariates included pre-pregnancy depression, gravida status, pet exposure, infant age at stool collection and breastfeeding status when investigating the effect of birth events on infant fecal SIgA level [32,37,38]. A directed acyclic graph (DAG) was generated with the daggity.net program to identify putative confounding factors of the relationship between birth events and infant fecal SIgA without introducing collider bias [39]. Applying the change-in-estimate procedure in linear regression to the minimum set of confounding variables from the DAG model [40], only gravida status was found to be a confounding factor for adjustment (Appendix A). Therefore, independence of causal pathway between birth events or breastfeeding status and fecal SIgA was assessed using mediation analysis, adjusting for gravida status (Appendix A). 

## 3. Results

In this large subsample of 1017 full-term infants from the CHILD general population birth cohort (Appendix A), there was an almost equal distribution of male (53.7%) and female infants. The majority of their mothers were of Caucasian ethnicity (75.4%), university-educated (55.3%), aged 30 to 39 years (63.2%) and multigravida (62.8%); most women did not smoke (94.5%) or experience perinatal depression (64.9%). A sizeable percentage of mothers did not have (45.4%) or had both pre- and postnatal pet exposure (45.9%). Over half of infants (53.2%) were born vaginally without maternal IAP exposure, 22.0% infants were delivered vaginally following IAP exposure, 10.4% and 14.4% of infants were born by CS delivery with and without labor, respectively. Almost half of infants were exclusively breastfed (48.3%), while 29.0% and 22.7% of infants were partially and formula-fed at 3 months of age, respectively. One third of infants who were breastfed over 12 months (37.5%), 39.8% and 22.7% of infants ceased breastfeeding between 3 and 12 months and <3 months, respectively. Antibiotic exposure rates were the lowest and exclusive breastfeeding rates the highest during the first 3 months of life in IAP-unexposed, vaginally delivered infants. Exclusive breastfeeding rates were lower following CS without labor than CS with labor. Maternal age, education level and gravida status differed by birth mode (Appendix A).

### 3.1. Birth Mode, SIgA Levels and Gut Microbiota of Infants at 3 and 12 Months

Infants born by CS with labor had a lower fecal SIgA level at 3 months compared to vaginal delivery without IAP exposure (4.85 vs. 6.27 mg/g feces, *p* < 0.05, Figure 1A); this difference was not observed at 1 year of age (Figure 1C). At 3 months, exclusive breastfeeding was associated with higher levels of SIgA (8.97 mg/g feces) than partial (5.58 mg/g feces, *p* < 0.0001, Figure 1B) or non-breastfeeding (2.71 mg/g feces, *p* < 0.0001, Figure 1B).

Birth events and breastfeeding exclusivity were major drivers of infant gut microbial community structure at 3 months (*p* = 0.001, Appendix A), but less so at 1 year (*p* < 0.1, Appendix A). Maternal IAP influenced infant gut microbial richness, especially at 3 months of age (*p* < 0.01, Appendix A). All three alpha-diversity indices were significantly lower among breastfed than non-breastfed infants at 3 months, with the lowest levels observed among exclusively breastfed infants (*p* < 0.01, Appendix A).

Consistent with results at the phylum level, genus *Bacteroides* of family Bacteroidaceae were under-represented at 3 months in CS-delivered and other IAP-exposed infants (Figure 2A,B). However, the family Clostridiaceae, Lachnospiraceae, and Veillonellaceae of phylum Firmicutes were enriched among infants exposed to maternal IAP during delivery (Figure 2A). IAP during vaginal birth was associated with reduced genus *Bifidobacterium* levels at 3 months of age but greater abundance of *Rothia* in the Actinobacteria phylum (Figure 2B). CS delivery, regardless of labor duration, was over-represented by members of the Proteobacteria, especially by the family Enterobacteriaceae, and unclassified Enterobacteriaceae, *Citrobacter*, and *Trabulsiella* at the genus level compared to vaginal birth without IAP (Figure 2A,B). Genus *Streptococcus* was also more abundant in 3-month old infants following CS delivery (Figure 2B). At 1 year of age, compared to vaginal birth without IAP, *Haemophilus* of phylum Proteobacteria and *Streptococcus* were more abundant among CS-delivered infants with and without labor, respectively; Bacteroidetes were depleted following CS no labor (Appendix A). Compared to infants born vaginally with no IAP, CS-delivered infants were more likely to be colonized by *C. difficile* 3 months later (Figure 3A), and this trend persisted to age 1 year among infants born by CS without labor (Appendix A).

At 3 months of age, exclusive breastfeeding was associated with elevated abundance of the Proteobacteria, specifically the Enterobacteriaceae family; genera *Haemophilus* and *Citrobacter* (Figure 2C,D); and genera from other phyla, such as *Lactobacillus*, *Bifidobacterium* and *Bacteroides* (Figure 2D). In contrast, members of Firmicutes phylum, such as genus *Streptococcus* and *Veillonella* were enriched in non-breastfed infants at this age (Figure 2D). *C. difficile* were more likely to colonize 3-month old infants who were partially or non-breastfed at 3 months, compared to those exclusively breastfed (Figure 3B). At 1 year of age, the Proteobacteria and Actinobacteria were more abundant in infants still breastfed at 12 months (Appendix A). At this age, no differences in *C. difficile* colonization was observed among infants by breastfeeding duration (Appendix A).

### 3.2. SIgA Correlations with Gut Microbial Taxa and Metabolites

Many more correlations between 3-month SIgA and gut microbiota or metabolites were evident in infants born vaginally-no IAP or those exclusively breastfed (FDR < 0.05, Appendix A). Stronger positive correlations with SIgA were observed for the Gammaproteobacteria, including unclassified Enterobacteriaceae and *Haemophilus*, and genus *Rothia*, and *Streptococcus*,as well as *Prevotella* and *Lactobacillus* also showed positive correlation with SIgA (FDR < 0.05, Appendix A). SIgA was positively correlated with several metabolites, especially 4-aminobutyrate (GABA), acetoin, choline, ethanol, fucose, galactose, glucose, lactate, 1,2-propanediol and pyruvate (FDR < 0.05, Appendix A). These SIgA-correlated metabolites were positively correlated with unclassified Enterobacteriaceae, *Haemophilus*, *Rothia*, *Lactobacillus* and *Streptococcus* (FDR < 0.05, Appendix A), whereas 1,2-propanediol, choline and fucose were negatively correlated with *C. difficile* colonization (FDR < 0.05, Appendix A). At 1 year, only genus *Haemophilus* was positively correlated with SIgA (FDR < 0.05, Appendix A) and choline, ethanol, fucose, galactose, glucose or methanol levels (Appendix A). All statistically significant correlations between SIgA, microbial genera or metabolites, or *C. difficile* (FDR < 0.05) were tested in mediation models (Appendix A).

### 3.3. Mediating Microbe—Metabolite Pathways from Birth Mode to SIgA Levels

Adjusting for gravida status, potential causal pathways between birth mode, breastfeeding status and fecal SIgA were evaluated using statistical sequential mediation methods, and all microbe-metabolite mediating pathways over the first year of life are shown in Figure 4 and Table 1 and Appendix A. The greatest mediating (indirect) effect among infants delivered by CS with labor was degree of breastfeeding and infant fecal levels of galactose (adjusted β-coefficient, −0.17; 95%CI: −0.34, −0.05) or fucose at 3 months of age (adjusted β-coefficient, −0.13, 95%CI: −0.35, −0.03) (Figure 5 and Table 1 and Appendix A). The direct effects of both paths from breast milk (adjusted β-coefficient for milk dose, 1.80, 95%CI: 1.17, 2.43) or galactose (adjusted β-coefficient for galactose; 0.22, 95%CI: 0.10, 0.33) to SIgA were also significant (*p* < 0.05, Figure 4 and Table 1 and Appendix A). GABA and choline were also milk-related mediators of indirect paths to reduced levels of SIgA at 3 months after birth in CS-delivered infants with labor (adjusted β-coefficient for GABA, −0.09, 95%CI: −0.23, −0.01; adjusted β-coefficient for choline, −0.10, 95%CI: −0.27, −0.00; Figure 4 and Table 1 and Appendix A). Among those metabolites, choline showed the strongest positive association with fecal SIgA levels (adjusted β-coefficient for the b^’^ path, 3.87, 95%CI: 2.15, 5.59; Figure 4 and Table 1 and Appendix A). However, milk-related galactose, fucose and GABA-mediated pathways were not found among infants born by CS delivery without labor (Appendix A).

Positive correlations between unclassified Enterobacteriaceae or *Streptococcus* and fecal metabolites were observed in sequential mediating pathways for enhanced SIgA production at 3 months of age, whereas *C. difficile*’s inverse association with metabolite levels was a characteristic of mediating pathways towards reduced SIgA levels following CS delivery (*p* < 0.05, Figure 4, and Table 1, Appendix A). Notably, mediation models also showed a direct effect of Enterobacteriaceae on SIgA, independent of metabolite levels (coefficients for b paths, *p* < 0.05, Table 1, Appendix A). For example, in the unclassified Enterobacteriaceae-pyruvate CS labor-SIgA model in Table 1, the coefficient for the b path was 0.64 (95%CI: 0.26–1.03). Lactate, pyruvate, choline and 1,2-propanediol were key metabolites in 3-month old infants that sequentially mediated the CS-SIgA association with Enterobacteriaceae and *C. difficile*, the latter of which was inversely correlated with many fecal metabolite levels (*p* < 0.05, Figure 4, and Table 1, Appendix A). Lactate was the only metabolite to sequentially mediate the CS-SIgA association with *Streptococcus* (*p* < 0.05, Figure 4, Table 1, Appendix A). 

The putative impact of infant IAP exposure in lowering SIgA levels at age 3 months was sequentially mediated through pathways of less breast milk and lower levels of milk or microbial metabolites compared to vaginal birth without IAP exposure (Figure 4 and Appendix A). In addition, genus *Rothia* and its metabolites, pyruvate, lactate, and to a lesser extent 1,2-propanediol, mediated the association between vaginal delivery with maternal IAP exposure and SIgA induction (*p* < 0.05, Figure 4 and Table 1 and Appendix A).

## 4. Discussion

In this general population of 1017 infants, we found medical intervention during birth to influence gut SIgA levels of young infants through pathways statistically mediated by microbes and metabolites. SIgA is a component of the intestinal microbiota-mucin barrier, critical to the exclusion of pathogenic microbes; it also binds to specific gut microbiota to stabilize its composition [41]. Whereas CS and maternal IAP are well-documented to cause gut microbial dysbiosis in young infants [42], we also observed SIgA changes in their presence. Cesarean section after labor was found to have the greatest negative impact on gut SIgA levels 3 months after birth (CS-labor vs. vaginal-no IAP, 4.85 vs. 6.27 mg/g feces, *p* < 0.05), while statistically significant differences was not seen for CS without labor. This association with lower infant fecal SIgA levels was found to be sequentially mediated through breast milk constituents or metabolites produced by microbiota from breast milk constituents. The greatest mediating effect was seen with degree of breast milk feeding and infant fecal levels of galactose (adjusted β-coefficient, −0.17; 95%CI: −0.34, −0.05) or fucose (adjusted β-coefficient, −0.13, 95%CI: −0.35, −0.03). Both sugars are available to infants as free molecules from microbiota-metabolized human milk oligosaccharides (HMO) [43]. They become carbohydrate residues of secretory IgA during the glycosylation process; when they are not available, IgA secretion by the intestine is lowered and its binding to microbiota is altered [41,44].

Our study provides evidence for the role of milk-related metabolites other than milk IgA in also stimulating gut SIgA levels in infants since statistical mediation or the indirect effect via the metabolite path (adjusted β-coefficient for galactose; 0.22, 95%CI: 0.10, 0.33) was found to be separate from the direct effect of breast milk on SIgA levels (adjusted β-coefficient for milk dose, 1.80, 95%CI: 1.17, 2.43). In this study, the direct effect was inferred to be SIgA supplied by breast milk, while indirect effects through microbes and metabolites were inferred to be pathways that stimulate endogenous SIgA production in infants. GABA was the other statistically significant milk-related mediator, attenuated by CS with labor. It is found in low levels in human milk but can be synthesized by lactobacilli, *E. coli* and *Bacteroides* sp. from glutamate, which is plentiful in milk [45,46,47]. GABA administration has been shown to stimulate SIgA levels in mice and humans [48,49]. Galactose, fucose and GABA metabolites statistically mediated the association between infant SIgA and CS with but not without labor, even though CS without labor resulted in the lowest breastfeeding rates. This finding suggests the involvement of hormones or other molecules released during labor in metabolite synthesis, HMO degradation or glycosylation of IgA [50]. 

Microbiota metabolites, lactate and pyruvate, statistically mediated the SIgA association with CS in the presence or absence of labor. They were in sequential mediation pathways that included infant gut Enterobacteriaceae, *C. difficile* or *Streptococcus* (only lactate). These findings suggest that gut microbiota affect metabolite availability and in turn, SIgA production. Lactate is produced by bifidobacteria, streptococci and enterobacteria [51]; feeding CS-delivered infants formula with a bifidobacteria probiotic raises fecal lactate, as well as SIgA levels [52]. Pyruvate is produced in microbiota manufacture of short-chain fatty acids and may promote the longevity of B cells to produce IgA [53]. Stool enrichment with streptococci or enterobacteria post CS with labor was positively correlated with higher lactate, and SIgA levels in our study infants. The mediation models also indicated a direct effect of Enterobacteriaceae on SIgA, independent of lactate or pyruvate. As shown with IgA-secreting plasma cells [54], we interpret these findings to indicate a stimulatory effect of enterobacteria on gut cell IgA secretion. On the other hand, *C. difficile* colonization following CS with labor or without labor lowered lactate and pyruvate, likely via metabolic consumption [55], and subsequently reduced SIgA levels. We posit that the balance of *C. difficile* (more likely with formula-feeding) versus enterobacteria (more likely with breastfeeding) post cesarean delivery determines fecal levels of lactate and pyruvate, and ultimately SIgA production.

Choline was another milk-related statistical mediator of the CS with or without labor-SIgA association. By far, it was the metabolite with the strongest direct association with fecal SIgA levels. Experimentally, choline deficiency leads to lowered fecal IgA levels [56]. Choline is present in breast milk and formula [57], and is released into the gut during the recycling of mucin and epithelial cell lipids, such as phosphatidylcholine [58]. An intact mucin barrier with phosphatidylcholine prevents SIgA breakdown by intestinal proteases [59]. Choline’s availability for the synthesis of phosphatidylcholine is essential in the antibody class switching process of plasma B cells to produce IgA [60]. Noteworthy is that B cell isotype switching and high serum SARS-CoV-2-specific IgA levels are characteristic of initial COVID-19 infection in adults [61] and infants [62]. Anti-SARS-CoV-2 IgA antibody responses are detected in the stool of patients with severe COVID-19 [63]. Choline levels in the gut lumen are ultimately determined by a limited set of choline-utilizing gamma and delta-Proteobacteria, and/or microbiota which compete with them [64]. In our study, choline sequentially mediated the CS-SIgA association with enterobacteria and *C. difficile*, the latter of which lowered fecal choline levels, as also shown by others [65]. We found a similar sequential mediation with these same gut microbiota and fecal 1,2-propanediol, which can be generated by microbial fermentation of fucose in HMOs or mucin [66]. In vitro addition of 1,2-propanediol has led to greater expansion and activity of intestinal B cells [67].

Genus *Rothia* of the Actinobacteria phylum were also identified as statistical mediators but of the path from infant exposure to maternal intrapartum antibiotics during vaginal birth to higher fecal SIgA levels. Unique from other mediating microbiota, *Rothia* are also found in oral microbiota, more commonly during formula-feeding [68]. Their fecal abundance is higher in infants with gastrointestinal disorders, which like formula, results in higher protein loads in the colon [69,70]. *Rothia* possess enzymes to break down complex dietary proteins such as gluten [71]. It is interesting that *Rothia* are found to be enriched in both oral and fecal microbiota of patients with COVID-19 infection [72]. In our study, *Rothia* metabolites, lactate and to a lesser extent, 1,2-propanediol [73,74], were found to be in the *Rothia* pathway to SIgA induction.

The CHILD Cohort Study, possessing a large, comprehensive and representative dataset, facilitated the thorough assessment of gut microbe-metabolite mediated pathways from birth mode to gut SIgA. We pioneered exploration of statistical sequential mediation between birth mode and SIgA levels in early life, the window of opportunity for gut microbiota modulation of immune development. The sequential mediation model allows mediators to be linked serially in a causal way rather than in parallel, which enabled testing of putative causal relationships between independent, multiple mediating and dependent variables. The large dataset included adequate sample size to ensure model adjustment for early life covariates in assessments of 3-month SIgA and metabolites. Sample size was not sufficient for assessment of SIgA at 12 months. Further, we lacked quantification data on breast milk SIgA, and thus, cannot exclude the influence of SIgA in mother’s milk. Our 16S rRNA gene sequencing approach also limited taxonomic resolution at the species and strain level. Finally, the biological plausibility of the reported mediating pathways requires experimental verification.

## 5. Conclusions

Through statistical mediation methods we identified candidate metabolic pathways involving galactose, fucose, GABA, choline, lactate, pyruvate and 1,2-propanediol to altered gut immunity in young infants following cesarean section delivery. When breastfeeding is less likely following cesarean, the reduced availability of these milk and microbiota metabolites leads to lowered SIgA levels in the infant gut. Hence, the benefit of breastfeeding on gut immunity following cesarean delivery is not only direct provision of SIgA to the nursing infant but also promotion of an infant’s own gut SIgA production through milk-related metabolites. Raising levels of metabolites, such as choline, has many additional benefits, including protection against impaired infant brain development in pregnancies complicated by viral infection [75].

## Figures and Tables

**Figure 1 metabolites-13-00148-f001:**
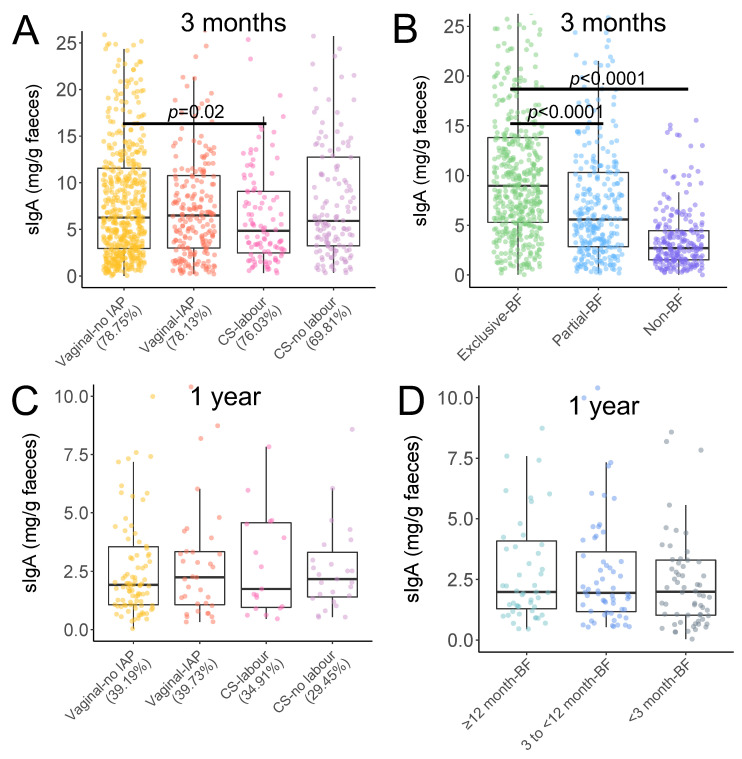
Infant fecal SIgA concentrations according to birth events (**A**,**C**) and breastfeeding status (**B**,**D**) at 3 months and 1 year. Following different birth event, the percentage of infants who were breastfed at 3 months (**A**) and those who were breastfed greater than or equal to 12 months at 1 year (**C**) are shown in the brackets. Significant differences were determined using crude linear regression model of the Box-Cox transformed values. Vaginally delivery without IAP exposure and exclusive breastfeeding are reference groups.

**Figure 2 metabolites-13-00148-f002:**
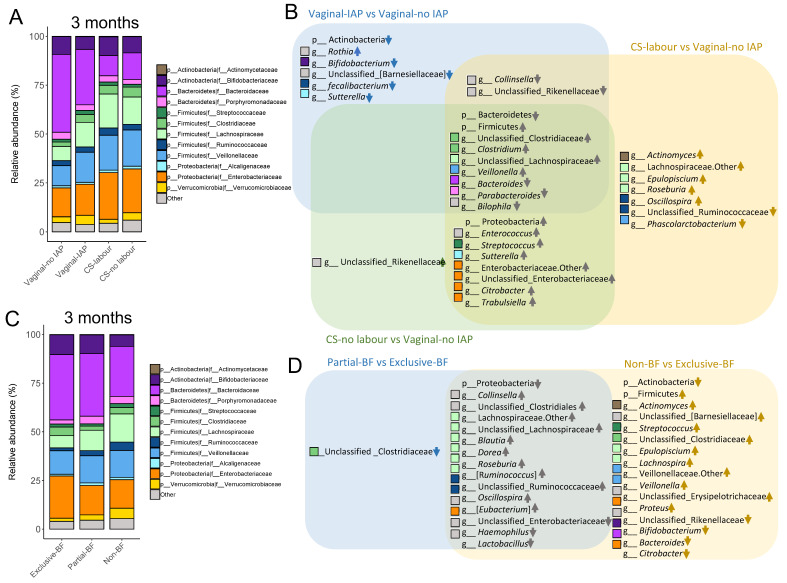
Specific microbial taxa in the infant gut are associated with birth events and breastfeeding exclusivity at 3 months. (**A**,**C**), average gut microbial taxa at family level of infants, showing the bacterial families with average relative abundance > 1%. (**B**,**D**), the discriminative taxonomic biomarkers of infant gut microbial taxa at phylum and genus level were identified using LEfSe. Significant differences were determined by an LDA score > 2 and *p*-value < 0.05. Upwards arrows indicate higher abundance, whereas downwards arrows indicate lower abundance of microbial taxa than reference groups (vaginal-no IAP and exclusive-BF for birth events and breastfeeding exclusivity, respectively). Arrows for shared taxa in overlapping area were in grey and arrows for unique taxa were in specific colors. Bacterial genera (**B**,**D**) and their corresponding family (**A**,**C**) share the same colors of square.

**Figure 3 metabolites-13-00148-f003:**
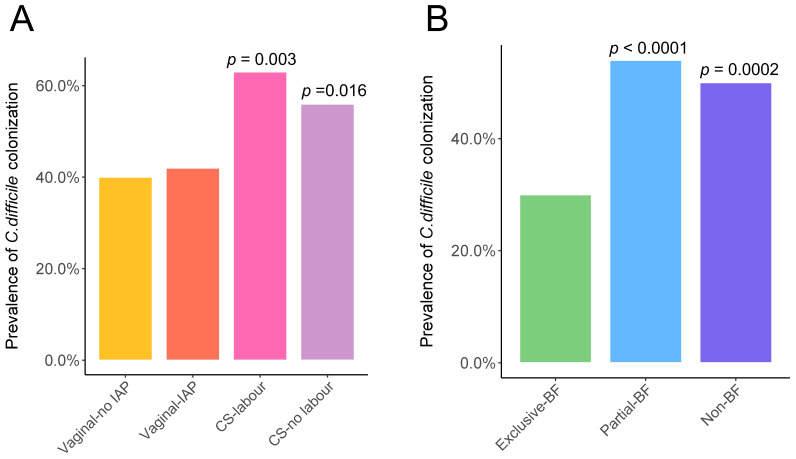
Prevalence of *C. difficile* colonization according to birth events (**A**) and breastfeeding exclusivity (**B**) at 3 months of age. Fisher’s exact tests with vaginal-no IAP or exclusive-BF as the reference group.

**Figure 4 metabolites-13-00148-f004:**
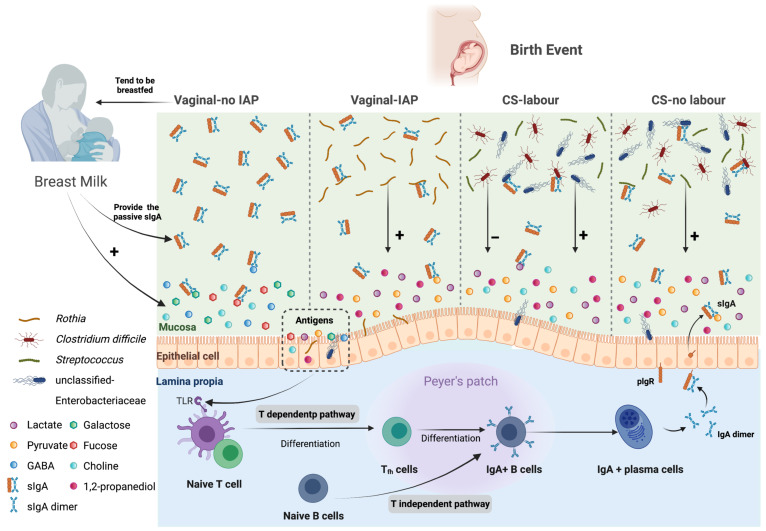
Putative microbe-metabolite mediated pathways from birth event to endogenous SIgA production among infants at 3 months of age, according to sequential mediation analysis, adjusted for gravida status. Compared to vaginal delivery without maternal IAP exposure, CS with labor is associated with lower level of SIgA. Infants born by vaginal delivery without IAP exposure tend to be breastfed and acquire passive SIgA, choline, GABA and lactate from breast milk. Lactate acts as the major putative microbial metabolite that stimulates endogenous SIgA production, via T cell-dependent and T cell-independent pathways. *Rothia*, Unclassified Enterobacteriaceae, and *Streptococcus* act independently or jointly with elevated lactate as the putative path for endogenous SIgA stimulation among vaginally delivered infants with IAP or those born by CS. The colonization of *Clostridium difficile* inhibits SIgA production by decreasing the level of lactate.

**Figure 5 metabolites-13-00148-f005:**
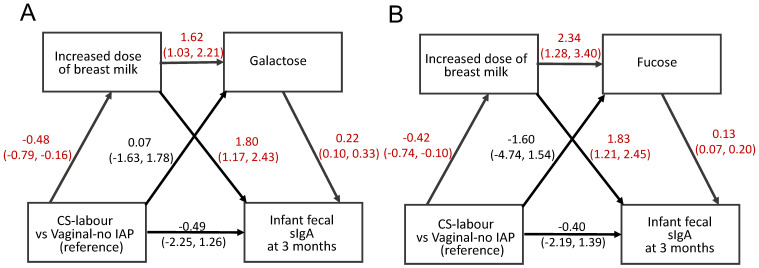
Causal diagram of the association between CS delivery with labor and infant fecal SIgA at 3 months of age, sequentially mediated by increased dose of breast milk, galactose (**A**) or fucose (**B**). Vaginal-no IAP is the reference group. Beta-coefficiency with 95% confidence interval of each path exhibits and significant differences (*p* < 0.05) are indicated in red.

**Table 1 metabolites-13-00148-t001:** The indirect effect of birth events on fecal SIgA level at 3 months through microbe-metabolite pathways, adjusted for gravida status ^†^.

			Total Indirect Effect			Pathway
Birth Event (X)	Mediator1 (M1)	Mediator2 (M2)	Beta-Coefficiency (95% CI)	Path	Mediator	Beta-Coefficiency (95% CI)
CS-labor	Increased dose of	Galactose	**−0.17 (−0.34, −0.05)**	a	M1	**−0.48 (−0.79, −0.16)**
	breast milk			a′	M2	0.07 (−1.63, 1.78)
				d	M1-M2	**1.62 (1.03, 2.21)**
				b	M1	**1.80 (1.17, 2.43)**
				b′	M2	**0.22 (0.10, 0.33)**
				c	CS-labor	−0.49 (−2.25, 1.26)
CS-labor	Increased dose of	Fucose	**−0.13 (−0.35, −0.03)**	a	M1	**−0.42 (−0.74, −0.10)**
	breast milk			a′	M2	−1.60 (−4.74, 1.54)
				d	M1-M2	**2.34 (1.28, 3.40)**
				b	M1	**1.83 (1.21, 2.45)**
				b′	M2	**0.13 (0.07, 0.20)**
				c	CS-labor	−0.40 (−2.19, 1.39)
CS-labor	Increased dose of	4Aminobutyrate	**−0.09 (−0.23, −0.01)**	a	M1	**−0.45 (−0.80, −0.10)**
	breast milk	(GABA)		a′	M2	−0.39 (−1.21, 0.42)
				d	M1-M2	**0.48 (0.21, 0.75)**
				b	M1	**2.03 (1.36, 2.69)**
				b′	M2	**0.41 (0.12, 0.69)**
				c	CS-labor	−0.87 (−2.83, 1.09)
CS-labor	Increased dose of	Choline	**−0.10 (−0.27, 0.00)**	a	M1	**−0.23 (−0.46, 0.00)**
	breast milk			a′	M2	0.00 (−0.09, 0.09)
				d	M1-M2	**0.11 (0.08, 0.15)**
				b	M1	**1.98 (1.29, 2.68)**
				b′	M2	**3.87 (2.15, 5.59)**
				c	CS-labor	−0.85 (−2.59, 0.88)
CS-labor	unclassified	Pyruvate	**0.20 (0.06, 0.45)**	a	M1	**0.64 (0.24, 1.04)**
	Enterobacteriaceae			a′	M2	−0.56 (−1.35, 0.23)
				d	M1-M2	**0.45 (0.28, 0.63)**
				b	M1	**0.64 (0.26, 1.03)**
				b′	M2	**0.68 (0.49, 0.87)**
				c	CS-labor	−1.60 (−3.32, 0.12)
CS-labor	unclassified	1,2-propanediol	**0.15 (0.04, 0.33)**	a	M1	**0.63 (0.21, 1.04)**
	Enterobacteriaceae			a′	M2	−1.43 (−3.09, 0.24)
				d	M1-M2	**0.93 (0.56, 1.30)**
				b	M1	**0.71 (0.22, 1.20)**
				b′	M2	**0.26 (0.14, 0.38)**
				c	CS-labor	−2.06 (−4.22, 0.10)
CS-labor	unclassified	Choline	**0.15 (0.04, 0.33)**	a	M1	**0.64 (0.24, 1.05)**
	Enterobacteriaceae			a′	M2	−0.05 (−0.15, 0.04)
				d	M1-M2	**0.05 (0.03, 0.07)**
				b	M1	**0.70 (0.30, 1.10)**
				b′	M2	**4.57 (2.84, 6.30)**
				c	CS-labor	−1.74 (−3.53, 0.04)
CS-labor	unclassified	Lactate	**0.09 (0.02, 0.19)**	a	M1	**0.60 (0.20, 0.99)**
	Enterobacteriaceae			a′	M2	−5.94 (−12.12, 0.24)
				d	M1-M2	**2.63 (1.26, 4.00)**
				b	M1	**0.78 (0.35, 1.22)**
				b′	M2	**0.06 (0.03, 0.09)**
				c	CS-labor	−1.77 (−3.70, 0.17)
CS-labor	*C. difficile*	1,2-propanediol	**−0.12 (−0.30, −0.03)**	a	M1	**4.16 (1.78, 6.53)**
				a′	M2	−0.09 (−1.96, 1.77)
				d	M1-M2	**−0.11 (−0.18, −0.03)**
				b	M1	−0.05 (−0.15, 0.04)
				b′	M2	**0.28 (0.16, 0.40)**
				c	CS-labor	−1.1 (−3.51, 1.31)
CS-labor	*C. difficile*	Pyruvate	**−0.12 (−0.31, −0.02)**	a	M1	**3.73 (1.44, 6.02)**
				a′	M2	0.05 (−0.87, 0.97)
				d	M1-M2	**−0.04 (−0.08, −0.01)**
				b	M1	−0.05 (−0.13, 0.03)
				b′	M2	**0.73 (0.53, 0.93)**
				c	CS-labor	−0.91 (−2.89, 1.08)
CS-labor	*C. difficile*	Choline	**−0.12 (−0.27, −0.02)**	a	M1	**3.52 (1.25, 5.80)**
				a′	M2	0.00 (−0.10, 0.11)
				d	M1-M2	**−0.01 (−0.01, 0.00)**
				b	M1	−0.04 (−0.12, 0.04)
				b′	M2	**5.20 (3.40, 7.00)**
				c	CS-labor	−0.99 (−3.03, 1.05)
CS-labor	*C. difficile*	Lactate	**−0.09 (−0.20, −0.01)**	a	M1	**3.10 (0.81, 5.38)**
				a′	M2	−1.74 (−8.70, 5.22)
				d	M1-M2	**−0.44 (−0.72, −0.17)**
				b	M1	−0.05 (−0.14, 0.04)
				b′	M2	**0.06 (0.03, 0.09)**
				c	CS-labor	−1.02 (−3.23, 1.19)
CS-labor	*Streptococcus*	Lactate	**0.09 (0.00, 0.23)**	a	M1	**0.47 (0.03, 0.91)**
				a′	M2	−5.85 (−11.94, 0.24)
				d	M1-M2	**3.13 (1.93, 4.33)**
				b	M1	0.20 (−0.19, 0.60)
				b′	M2	**0.06 (0.04, 0.09)**
				c	CS-labor	−1.37 (−3.32, 0.58)
CS-no labor	unclassified	Pyruvate	**0.15 (0.01, 0.39)**	a	M1	**0.48 (0.10, 0.86)**
	Enterobacteriaceae			a′	M2	−0.61 (−1.36, 0.15)
				d	M1-M2	**0.45 (0.28, 0.63)**
				b	M1	**0.64 (0.26, 1.03)**
				b′	M2	**0.68 (0.49, 0.87)**
				c	CS-no labor	0.29 (−1.35, 1.93)
CS-no labor	unclassified	Choline	**0.11 (0.01, 0.26)**	a	M1	**0.46 (0.08, 0.84)**
	Enterobacteriaceae			a′	M2	−0.08 (−0.16, 0.01)
				d	M1-M2	**0.05 (0.03, 0.07)**
				b	M1	**0.70 (0.30, 1.10)**
				b′	M2	**4.57 (2.84, 6.30)**
				c	CS-no labor	0.27 (−1.41, 1.95)
CS-no labor	unclassified	1,2-propanediol	**0.09 (0.01, 0.26)**	a	M1	**0.40 (0.04, 0.76)**
	Enterobacteriaceae			a′	M2	−1.38 (−2.84, 0.08)
				d	M1-M2	**0.93 (0.56, 1.30)**
				b	M1	**0.71 (0.22, 1.20)**
				b′	M2	**0.26 (0.14, 0.38)**
				c	CS-no labor	0.14 (−1.75, 2.03)
CS-no labor	unclassified	Lactate	**0.07 (0.01, 0.16)**	a	M1	**0.46 (0.10, 0.81)**
	Enterobacteriaceae			a′	M2	**−6.22 (−11.82, −0.61)**
				d	M1-M2	**2.63 (1.26, 4.00)**
				b	M1	**0.78 (0.35, 1.22)**
				b′	M2	**0.06 (0.03, 0.09)**
				c	CS-no labor	−0.22 (−1.98, 1.53)
CS-no labor	*C. difficile*	1,2-propanediol	**−0.11 (−0.26, −0.02)**	a	M1	**3.68 (1.66, 5.71)**
				a′	M2	−0.44 (−2.03, 1.15)
				d	M1-M2	**−0.11 (−0.18, −0.03)**
				b	M1	−0.05 (−0.15, 0.04)
				b′	M2	**0.28 (0.16, 0.40)**
				c	CS-no labor	0.58 (−1.47, 2.64)
CS-no labor	*C. difficile*	Pyruvate	**−0.11 (−0.28, −0.01)**	a	M1	**3.38 (1.30, 5.45)**
				a′	M2	−0.30 (−1.14, 0.54)
				d	M1-M2	**−0.04 (−0.08, −0.01)**
				b	M1	−0.05 (−0.13, 0.03)
				b′	M2	**0.73 (0.53, 0.93)**
				c	CS-no labor	0.84 (−0.97, 2.64)
CS-no labor	*C. difficile*	Choline	**−0.11 (−0.26, −0.02)**	a	M1	**3.36 (1.31, 5.42)**
				a′	M2	−0.02 (−0.12, 0.07)
				d	M1-M2	**−0.01 (−0.01, 0.00)**
				b	M1	−0.04 (−0.12, 0.04)
				b′	M2	**5.20 (3.40, 7.00)**
				c	CS-no labor	0.76 (−1.08, 2.61)
CS-no labor	*C. difficile*	Lactate	**−0.09 (−0.19, −0.02)**	a	M1	**3.17 (1.20, 5.15)**
				a′	M2	−3.29 (−9.33, 2.76)
				d	M1-M2	**−0.44 (−0.72, −0.17)**
				b	M1	−0.05 (−0.14, 0.04)
				b′	M2	**0.06 (0.03, 0.09)**
				c	CS-no labor	0.27 (−1.65, 2.19)
CS-no labor	*Streptococcus*	Lactate	**0.11 (0.04, 0.24)**	a	M1	**0.58 (0.17, 0.98)**
				a′	M2	**−6.82 (−12.36, −1.27)**
				d	M1-M2	**3.13 (1.93, 4.33)**
				b	M1	0.20 (−0.19, 0.60)
				b′	M2	**0.06 (0.04, 0.09)**
				c	CS-no labor	0.04 (−1.74, 1.82)
Vaginal-IAP	*Rothia*	Pyruvate	**0.12 (0.01, 0.33)**	a	M1	**0.77 (0.09, 1.45)**
				a′	M2	−0.45 (−1.09, 0.19)
				d	M1-M2	**0.23 (0.15, 0.32)**
				b	M1	**0.28 (0.09, 0.46)**
				b′	M2	**0.68 (0.49, 0.88)**
				c	Vaginal-IAP	−0.51 (−1.91, 0.89)
Vaginal-IAP	*Rothia*	Lactate	**0.08 (0.01, 0.17)**	a	M1	**0.77 (0.08, 1.46)**
				a′	M2	**−6.75 (−11.74, −1.76)**
				d	M1-M2	**1.76 (1.13, 2.38)**
				b	M1	**0.35 (0.14, 0.55)**
				b′	M2	**0.06 (0.03, 0.08)**
				c	Vaginal-IAP	−0.56 (−2.15, 1.04)
Vaginal-IAP	*Rothia*	1,2-propanediol	**0.05 (0.00, 0.15)**	a	M1	**0.72 (−0.01, 1.45)**
				a′	M2	−0.58 (−1.93, 0.78)
				d	M1-M2	**0.26 (0.09, 0.44)**
				b	M1	**0.38 (0.16, 0.60)**
				b′	M2	**0.27 (0.15, 0.38)**
				c	Vaginal-IAP	−0.83 (−2.55, 0.89)

^†^ Sequential mediation analysis is conducted using vaginal-no IAP as the reference group. Detailed information regarding causal relationships of each path (a, a′, b, b′, c, and d) for CS delivery with or without labor, vaginal delivery with IAP (X), mediator1 (M1) and mediator2 (M2) are indicated in Appendix A. Significant difference (*p* < 0.05) are indicated in bold. Total indirect effect indicates the path a→d→b′ in the sequential mediation model.

## Data Availability

Restrictions apply to the availability of these data. They were obtained from the CHILD Cohort Study and are available via childcohort.ca (http://childcohort.ca/ (accessed on 1 April 2022)) with the permission of Anita Kozyrskyj and the Child Cohort Study National Coordinating Centre.

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
