# Peer review of "Impact of Cesarean Delivery and Breastfeeding on Secretory Immunoglobulin A in the Infant Gut Is Mediated by Gut Microbiota and Metabolites"

_metabolites, 2023, doi:10.3390/metabo13020148_

Round 1

Reviewer 1 Report

This study investigates the impact of Caesarean delivery on faecal secretory IgA in relation to gut microbiota and metabolites at 3 and 12 months of age. While novel and interesting findings are presented, the conclusions that effects on faecal SIgA levels were directly mediated by gut microbiota and metabolites need to be toned down, as the findings of the sequential mediation model require experimental verification. Furthermore, the limitations of the sequential mediation analyses need to be addressed. 

Comments

1.     Abstract: To not confuse secretory and serum IgA, SIgA is a preferrable abbreviation to sIgA (to be revised throughout the manuscript).

2.     Abstract: It should be clarified that IgA and metabolite levels were only available from 175 samples at 12 months.

3.     Abstract, Results and Conclusion: The conclusions that effects on faecal SIgA levels were directly mediated by gut microbiota and metabolites need to be toned down, as the findings of the sequential mediation model require experimental verification.

4.     Introduction: Bifidobacterium abundance is not presented in ref 24, and presence or absence of labour in CS delivered infants was not associated with Bifidobacterium abundance in ref 23.

5.     Introduction, line 68: To tone down the conclusion, please change “we identified mediating” to “we identified potentially mediating”.

6.     Methods, line 107: Why was IgA data available for only 175 infants at 12 months?

7.     Methods, line 108: In how many samples were metabolite concentrations determined at 3 and 12 months? These numbers should be stated.

8.     Methods, line 118: PD should be defined (phylogenetic diversity).

9.     Methods, line 158: In the legend to Fig S2, “level” should be changed to “levels”.

10.  Results, first paragraph: It should be mentioned that maternal age and parity differed between the birth mode groups.

11.  Results, Fig S3A and S3C: The Vaginal-IAP and CS-labour groups are difficult to distinguish; more contrasting colours should be used.

12.  Results, line 196: It should be stated that Bacteroides abundance was reduced in CS infants also.

13.  Results, line 197: Vellionellaceae are not found in Fig 2A.

14.  Results, line 207: The reduced Bacteroidetes abundance in CS-no labour vs vaginal-no IAP should be mentioned also.

15.  Results, line 233: Proteobacteria and Actinobacteria were differently abundant in relation to BF status, but not the most highly abundant phyla.

16.  Results, line 239: These correlations are shown in Fig S7, S8 and S10 only, please correct.

17.  Results, line 239: How were strong correlations defined? The Rho values seem to be mostly below 0.3 and thus quite weak.

18.  Results, line 241: Strong correlations with Prevotella and Lactobacillus are not presented in Fig S7.

19.  Results, line 243: Since 1-2-propanediol (an e should be added to “propandiol)” is used in the Results and Discussion, this name should be used in the figures and tables also, instead of propyleneglycol.

20.  Results, line 245: All the metabolites on line 242-243 were not negatively correlated with C difficile colonisation; this needs to be corrected,

21.  Results, line 248: It should be explained that correlations with SIgA are shown in Fig S12 and S13.

22.  Results, line 255: Table 1 is not well formatted.

23.  Results, line 261-262: This sentence is unclear and need to be rephrased.

24.  Results, line 288: This effect is not shown in Table 1, nor in Table 2.

25.  Results, line 297 and line 300: These findings are not shown in Table S2.

26.  Discussion: The conclusions that effects on faecal SIgA levels were directly mediated by gut microbiota and metabolites need to be toned down in several sections in the Discussion, as the findings of the sequential mediation model require experimental verification. The limitations of the sequential mediation analyses need to be addressed.

27.  Discussion, line 317: What could be the reason for the absence of differences for CS without labour?

28.  Discussion, line 319: The meaning of “microbiota from these constituents” is unclear.

29.  Discussion, line 343-345: This sentence is unclear and need to be rephrased.

30.  Discussion, line 362: It is not clear that choline showed the strongest positive correlation by far (in Fig S8).

31.  Discussion, line 400: That data were available from only 175 samples at 12 months was another limitation that should be mentioned.

Author Response

Comments

  1. Abstract: To not confuse secretory and serum IgA, SIgA is a preferrable abbreviation to sIgA (to be revised throughout the manuscript).

 We agree. The acronym was revised throughout the paper as requested.

  1. Abstract: It should be clarified that IgA and metabolite levels were only available from 175 samples at 12 months.

Since there were no birth mode differences in fecal SIgA at 12 months, we only highlighted findings for 3-month SIgA in the abstract. As such, we have added the N (of 551) for the 3-month SIgA mediation results to the abstract.

  1. Abstract, Results and Conclusion: The conclusions that effects on faecal SIgA levels were directly mediated by gut microbiota and metabolites need to be toned down, as the findings of the sequential mediation model require experimental verification.

In the Abstract and Discussion, we have now recommended experimental verification.

  1. Introduction: Bifidobacteriumabundance is not presented in ref 24, and presence or absence of labour in CS delivered infants was not associated with Bifidobacterium abundance in ref 23.

We deleted reference #24 (Thilaganathan et al) since it did not report on Bifidobacterium; it was accidentally left after the sentence was edited. Reference #23 is our own paper, in which differences in Bifidobacterium quantity but not abundance were found for CS with/without labour. We loosely and incorrectly stated ‘abundance’ in the sentence, which we have now corrected to ‘quantity.’

  1. Introduction, line 68: To tone down the conclusion, please change “we identified mediating” to “we identified potentially mediating”.

Implemented as requested.

  1. Methods, line 107: Why was IgA data available for only 175 infants at 12 months?

This is because there were no further funds to assay the remaining samples.

  1. Methods, line 108: In how many samples were metabolite concentrations determined at 3 and 12 months? These numbers should be stated.

These numbers were reported as 551 and 175, respectively.

  1. Methods, line 118: PD should be defined (phylogenetic diversity).

PD has been now been defined.

  1. Methods, line 158: In the legend to Fig S2, “level” should be changed to “levels”.

This edit has been made.

  1. Results, first paragraph: It should be mentioned that maternal age and parity differed between the birth mode groups.

Implemented as requested.

  1. Results, Fig S3A and S3C: The Vaginal-IAP and CS-labour groups are difficult to distinguish; more contrasting colours should be used.

This edit has been made.

  1. Results, line 196: It should be stated that Bacteroidesabundance was reduced in CS infants also.

Lines 82-84 originally stated that “Women colonized with vaginal Group B Streptococcus (GBS) or delivered by caesarean section were administered intrapartum penicillin or cefazolin, respectively.” These are standards of care. Thus, IAP-exposed infants included those delivered by CS. In case this connection is not made by readers, we edited the sentence as follows: Consistent with results at the phylum level, genus Bacteroides of family Bacteroidaceae were under-represented at 3 months in CS-delivered and other IAP-exposed infants.

  1. Results, line 197: Vellionellaceae are not found in Fig 2A.

Veillonellaceae is shown in Fig 2A with blue colour

  1. Results, line 207: The reduced Bacteroidetes abundance in CS-no labour vsvaginal-no IAP should be mentioned also.

This finding has been now been mentioned.

  1. Results, line 233: Proteobacteria and Actinobacteria were differently abundant in relation to BF status, but not the most highly abundant phyla.

The sentence has been revised accordingly.

  1. Results, line 239: These correlations are shown in Fig S7, S8 and S10 only, please correct.

The figure numbers have been corrected.

  1. Results, line 239: How were strong correlations defined? The Rhovalues seem to be mostly below 0.3 and thus quite weak.

As per the original intent, the adjective ‘strong’ has been corrected to ‘stronger’ to indicate ‘more strong.’

  1. Results, line 241: Strong correlations with Prevotellaand Lactobacillus are not presented in Fig S7.

The descriptor ‘strong correlation’ has been changed to ‘positive correlation.’

  1. Results, line 243: Since 1-2-propanediol (an e should be added to “propandiol)” is used in the Results and Discussion, this name should be used in the figures and tables also, instead of propyleneglycol.

It has been revised in all tables and figures.  

  1. Results, line 245: All the metabolites on line 242-243 were not negatively correlated with C difficilecolonisation; this needs to be corrected,

The sentence has been revised to “whereas 1,2-propanediol, choline and fucose were negatively correlated with C. difficile colonization.”

  1. Results, line 248: It should be explained that correlations with SIgA are shown in Fig S12 and S13.

Reference has now been made to these figures.

  1. Results, line 255: Table 1 is not well formatted.

We appreciate you noting this. It has been reformatted to improve its clarity.

  1. Results, line 261-262: This sentence is unclear and need to be rephrased.

We have corrected the grammar in the sentence (now lines 267-270), which should improve its clarity as follows: GABA and choline were also milk-related mediators of indirect paths to reduced levels of SIgA at 3 months after birth in CS-delivered infants with labour (adjusted β-coefficient for GABA, -0.09, 95%CI: -0.23, -0.01; adjusted β-coefficient for choline, -0.10, 95%CI: -0.27, -0.00; Figure 4 and Table 1&S2).

  1. Results, line 288: This effect is not shown in Table 1, nor in Table 2.

The effect is shown as the coefficient for path b in the Enterobacteriaceae mediation models. To make this more clear, we have added ‘coefficients for b paths’ in the parentheses and provided an example as follows: For example, in the unclassified Enterobacteriaceae-pyruvate CS labour-IgA model in Table 1, the coefficient for the b path was 0.64 (95%CI: 0.26-1.03).

There is no Table 2 in the manuscript. Likely the reviewer meant Table S2, in which b path coefficients are also reported.

  1. Results, line 297 and line 300: These findings are not shown in Table S2.

Thank you for catching this typo. We have deleted reference to Table S2 in these sentences.

  1. Discussion: The conclusions that effects on faecal SIgA levels were directly mediated by gut microbiota and metabolites need to be toned down in several sections in the Discussion, as the findings of the sequential mediation model require experimental verification. The limitations of the sequential mediation analyses need to be addressed.

We have reworded these sentences in the Discussion to state ‘statistically mediated,’ which is the precise term for the findings of our statistical analyses and should do the trick. We have also noted the need for experimental verification in lines 413-414.

  1. Discussion, line 317: What could be the reason for the absence of differences for CS without labour?

We found this lack of difference puzzling as well, because CS without labour resulted in the lowest breastfeeding rates. In the first paragraph where line 328 was found, we merely reported this result, along with other main findings. Later at the end of paragraph 2 in lines 351-353, we offer this explanation: This finding suggests the involvement of hormones or other molecules released during labour in metabolite synthesis, HMO degradation or glycosylation of IgA [50].

  1. Discussion, line 319: The meaning of “microbiota from these constituents” is unclear.

It refers to breast milk constituents. We have now specified this in the sentence.

  1. Discussion, line 343-345: This sentence is unclear and need to be rephrased.

This sentence has been split into 2 sentences to improve clarity as follows: Microbiota metabolites, lactate and pyruvate, statistically mediated the SIgA association with CS, in the presence or absence of labour. They were in sequential mediation pathways that included infant gut Enterobacteriaceae or C. difficile, or Streptococcus (only lactate).

  1. Discussion, line 362: It is not clear that choline showed the strongest positive correlation by far (in Fig S8).

The statement in former line 362 was in reference to path coefficients for choline and other metabolites, reported in mediation models in Tables 1 & S2 and highlighted in the Results section. Figure S8 reports crude correlations between metabolites and sIgA. To avoid confusion with Figure S8 crude correlations, we have replaced ‘correlation’ with ‘association’ in line 373 (formerly line 362) in the Discussion and lines 270-272 in the Results section; to the latter we had also specified ‘b’ coefficient.’

  1. Discussion, line 400: That data were available from only 175 samples at 12 months was another limitation that should be mentioned.

Inadequate sample size for SIgA at 12 months was added as a limitation in the Discussion.

Reviewer 2 Report

I think your article is very good analysis about the impact of birth mode and breastfeeding status on sIgA. The components of breast milk, including sIgA, are affected by a variety of factors. Therefore, it is not easy to control and analyze such factors. Also, due to the nature of the human milk, it is not easy to collect samples depending on the time. Nevertheless, the researchers seem to have made good progress on the difficult research. Please check only a few minor things.

1. Didn't the delivery mode and feeding type affect each other? At least it would be better to present that there is no statistical difference between each Delivery Mode and Feeding Pattern.

2. Personally, I think Figure 5 will be of great help to the reader. However, can table 1 be modified a little for the reader's readability and understanding?

Author Response

We thank you for your compliments. 

As shown in Figure S1, breastfeeding differences by birth mode were seen but they did not reach statistical significance. Of potential relevance to the results is the lower rate for exclusive breastfeeding in infants delivered by CS without labour than CS with labour. We have now noted this in the Results section.

Table 1 has been reformatted to aid readability.

Reviewer 3 Report

The manuscript reports an interesting and well described work. It describes the correlation between gut microbiome composition and birth events, breast or not feeding, intrapartum antibiotic prophylaxis, besides the circulating IgA levels related to partum and postpartum conditions. Conclusions are the summary of collected data.

I ask the authors only to improve the methods description, no details are available about the sequencing method and also metataxonomic analysis lacks of employed package names.

Author Response

We thank you for your compliments, and have now added details on the sequencing method and taxonomic analysis package name.

Round 2

Reviewer 1 Report

The authors have improved the paper and addressed the comments appropriately.